# Sequential phenotypic constraints on social information use in wild baboons

**Alecia J Carter[1]\*, Miquel Torrents Ticó[2], Guy Cowlishaw[3]**

[1]Department of Zoology, University of Cambridge, Cambridge, United Kingdom; [2]Zoological Society of London, Tsaobis Baboon Project, Institute of Zoology, London, United Kingdom; [3]Zoological Society of London, Institute of Zoology, London, United Kingdom

**Abstract** Social information allows the rapid dissemination of novel information among individuals. However, an individual's ability to use information is likely to be dependent on phenotypic constraints operating at three successive steps: acquisition, application, and exploitation. We tested this novel framework by quantifying the sequential process of social information use with experimental food patches in wild baboons (*Papio ursinus*). We identified phenotypic constraints at each step of the information use sequence: peripheral individuals in the proximity network were less likely to acquire and apply social information, while subordinate females were less likely to exploit it successfully. Social bonds and personality also played a limiting role along the sequence. As a result of these constraints, the average individual only acquired and exploited social information on <25% and <5% of occasions. Our study highlights the sequential nature of information use and the fundamental importance of phenotypic constraints on this sequence.

## Introduction

Individuals require information to reduce uncertainty about their environment. Information can be acquired in two ways (*Dall et al., 2005*): by interacting with the environment directly (personal information) or by attending to the behaviour of others (social information). Individuals benefit from both personal and social information in myriad contexts, including foraging, predator avoidance, and mate choice (*Giraldeau et al., 2002*). Their use is moderated by their expense and reliability: personal information is usually reliable but costly and time consuming to collect, social information is less costly but more likely to become outdated and unreliable (*Giraldeau et al., 2002*; *Laland, 2004*). The low costs of social information also allow it to disseminate more rapidly across groups, such that it can play an important role in the formation of traditions and cultures (*Whiten, 2000*; *Castro and Toro, 2004*).

Despite the fitness benefits of information use, we have very little understanding of how individuals vary in their ability to capture these benefits. Indeed, theory developed to explain the costs and benefits of information use usually assumes homogeneity within a population (e.g. *Pradhan et al., 2012*). To understand individual variation in information use, either personal or social, we suggest it is helpful to decompose the process into a sequence of three steps: the acquisition of information, its application, and the exploitation of its benefits (*Table 1*). Up until now, many studies have implicitly assumed that these three steps are synonymous, but recent evidence indicates that information use is substantially more complex. In particular, *Carter et al. (2014)* found that the time spent acquiring social information about a task did not correlate with subsequent performance (information application) in wild baboons (*Papio ursinus*), while *Atton et al. (2012)* found differences in individual performance between task discovery (information acquisition) and task solving (information

\*For correspondence: ac854@cam.ac.uk

**eLife digest** Animals need information to make decisions, and a quick way to get this information is to watch what others are doing. Animals, like humans, have different social networks that they could acquire this kind of 'social information' from, yet we know little about which networks they actually use. In addition, once an animal has obtained social information, some aspect of their lives, such as their sex or social rank, could prevent them from using it. Once again, however, we know very little about the impact of these personal constraints.

Carter et al. found that information about the location of a highly preferred food flowed through a social network of wild baboons that was based on who was regularly in close proximity to whom. However, while individuals with more neighbours were better at obtaining social information about food location, they were not better at using it. Rather, individuals were more likely to successfully exploit such information if they were dominant, bold, male, and had good social bonds with others.

Carter et al.'s results show that the use of social information is a process with several stages – from information acquisition, to its application, and finally its exploitation. Furthermore, the characteristics of an individual can limit their success at each of these stages. The next step is to figure out whether different types of social information – whether short- or long-lived, easy to acquire or more complex – flow through the same networks and have the same personal constraints on who can use them.

exploitation) in three-spine sticklebacks (*Gasterosteus aculeatus*). The recognition of three sequential steps allows us to begin unpacking the complexity of information use, and to explore variation in the performance of different individuals at different points along the sequence. Distinct sensory and motor capabilities are likely to be involved at each stage, leading to different phenotypic constraints. As a result, individuals who are effective at one step may be less so at another, with significant implications for who captures the most benefits.

The range of phenotypic constraints operating at each step might include cognitive, social, behavioural, ecological and demographic characteristics. The importance of these constraints is likely to differ not only between individuals but also between populations and species. Here, we focus on social, behavioural and demographic constraints on the social information use sequence. To begin with, we consider the social phenotype, i.e., phenotypic traits that emerge from social interactions with others and are likely to be under selection, in this case individual dominance rank (*Moore, 1993*) and position in the social network (*Aplin et al., 2015*). Dominant animals can aggressively monopolise resources such as mates (*Cowlishaw and Dunbar, 1991*) and food (*Koenig, 2002*), limiting the opportunities for others to apply and exploit information that they have acquired either personally or socially about these resources. This can further lead to voluntary inhibition in the use of information by subordinate animals, e.g., low-ranked rhesus monkeys (*Macaca mulatta*) only performed a socially-learnt task when high-ranked monkeys were not present (*Drea and Wallen, 1999*). The social network will likely manifest constraints on different stages of the information use sequence depending on the type of association indexed by the network, i.e., associations according to spatio-temporal proximity or direct interactions. For instance, positions in proximity networks may affect an individual's opportunities for information acquisition, assuming individuals are more likely to acquire information from others with whom they are more frequently in visual contact (*Coussi-Korbel and Fragaszy, 1995*; *Voelkl and Noë, 2008, 2010*), e.g., stickleback proximity networks predict the flow of information about the location of a novel task (*Atton et al., 2012*). Similarly, positions in interaction networks may limit the application and exploitation of information about resources if social bonds are required to gain access to those resources (*Henzi and Barrett, 2002*; *Clarke et al., 2010*), e.g., vervet monkeys (*Chlorocebus aethiops*) allocate their social effort to access food provided by others (*Fruteau et al., 2009*).

The behavioural phenotype, specifically personality, may also be important in mediating individuals' acquisition, application and exploitation of social information. We have previously shown that personality can affect both the first and second steps of the social information use sequence: calmer baboons were more likely to acquire social information but bolder individuals were more likely to

**Table 1.** The information use sequence: definitions and examples.

| Stage | Definition | Example(s) of stage |
|---|---|---|
| Acquisition | An individual gains knowledge | 1. Gaining knowledge of the location of a food patch.<br>2. Gaining knowledge of the location or form of a novel task. |
| Application | An individual uses the information that it has acquired in a relevant (but not necessarily successful) way | 1. Entering a food patch. Because information can become outdated, 'application' can occur even after the patch has been fully depleted, leading to no reward.<br>2. Using stimulus or local enhancement to manipulate a novel task, but not necessarily successfully. |
| Exploitation | An individual successfully uses information that it has acquired and applied to gain a benefit | 1. Gaining food from a patch.<br>2. Solving a novel task. |

apply it (*Carter et al., 2014*). In geese (*Branta leucopsis*), personality affected the final step in the sequence: shyer geese were more likely to exploit social information to forage where other geese were successfully foraging (*Kurvers et al., 2010*). Similarly, fast exploring great tits (*Parus major*) were more likely to apply social information and change their foraging behaviour to mirror a demonstrator's (*Marchetti and Drent, 2000*).

Finally, individual demographic characteristics, particularly age and sex, may affect each step of the social information use sequence. Juveniles may be more reliant on social information because adults have already acquired the necessary information to survive to adulthood (*Galef and Laland, 2005*). This prediction is supported in baboons, where juveniles spend more time than adults acquiring social information about a novel food (*Carter et al., 2014*). Similarly, in Japanese macaques (*Macaca fuscata*), novel socially-transmitted behaviours were more likely to be adopted by juveniles (*Huffman et al., 1996*). Sex differences in the social information use sequence are also likely due to, e.g., sex-specific costs of competition at resources (e.g. *Aragón, 2009*).

A further challenge involved in elucidating the social information use sequence is the identification of the relevant network through which information diffuses during the acquisition phase. Researchers have usually assumed information transfers primarily between individuals who are in close spatial proximity (for examples, see *Kendal et al., 2010*; *Aplin et al., 2012*; *Claidiére et al., 2013*). However, individuals may preferentially acquire information from others besides those to whom they are associated as neighbours. For instance, individuals may be more attentive to those with whom they have strong affiliative bonds (*Coussi-Korbel and Fragaszy, 1995*) or to lower ranking animals from whom they can scrounge resources (*King et al., 2009*). The need to consider alternative networks in the identification of information diffusion paths is well illustrated by *Boogert et al. (2014)*, who showed that the spread of solutions to a novel foraging task in captive starlings (*Sturnus vulgaris*) was better predicted by a network based on perching associations than foraging associations.

In this study, we explore phenotypic limitations on social information use. We examined information transmission among wild chacma baboons (*Papio ursinus*) by experimentally introducing ephemeral patches of a highly preferred food while the troops foraged naturally. We first compared which of five networks best predicted the diffusion of information through the troops about the location of a highly preferred food. Next, we investigated how individuals' social, behavioural and demographic phenotypes affected their abilities to successfully acquire, apply and exploit this social information.

## Materials and methods

### Study area and study species

We studied two habituated troops (J, L) of wild chacma baboons at Tsaobis Nature Park, Namibia (15° 45'E, 22° 23'S) from May to July 2014. Two habitat types make up the Tsaobis terrain: open desert and riparian woodland. The open desert is characterised by small herbs and shrubs, such as *Monechma cleomoides*, *Sesamum capense*, and *Commiphora virgata*, in a mosaic of alluvial plains and steep-sided hills surrounding the ephemeral Swakop River. The riparian woodland along the Swakop is characterised by large trees and bushes, such as *Faidherbia albida*, *Prosopis glandulosa*, and *Salvadora persica* (see *Cowlishaw and Davies, 1997* for more details). The baboons' diet largely

consists of berries, flowers, seedpods, and immature leaves (*Cowlishaw, 1997*). The baboons' main predator, the leopard (*Panthera pardus*), is rare at Tsaobis and the risk of predation is low.

The baboon troops were followed daily from dawn until dusk. We collected data on all baboons over 2 years of age, who were individually recognisable by marks (ear notches) ($N_J = 46$, $N_{J\ adult\ female} = 18$, $N_{J\ adult\ male} = 8$, $N_{J\ juvenile\ female} = 6$, $N_{J\ juvenile\ male} = 14$; $N_L = 48$, $N_{L\ adult\ female} = 19$, $N_{L\ adult\ male} = 10$, $N_{L\ juvenile\ female} = 2$, $N_{J\ juvenile\ male} = 17$). Individuals younger than 2 years did not have marks, were not individually recognisable and did not form part of the study. Dominance ranks were assessed through aggressive interactions, recorded *ad libitum*, using Matman 1.1.4 (Noldus Information Technology 2003). These data included all displacements, supplants, threats, chases and attacks that occurred for which we could identify both the actor and recipient. If more than one dominance behaviour occurred in one event, such as a threat followed by a chase, only one interaction was recorded. The dominance hierarchies were strongly linear (Landau's corrected linearity index: $h'_{J\ troop} = 0.162$, $h'_{L\ troop} = 0.183$, $N_J = 618$, $N_L = 856$, p<0.001 in both cases). Dominance rank was expressed relatively (which controls for group size), using the formula 1-[(1-$r$)/(1-$n$)] where $r$ is the individual's absolute rank and $n$ is the group size, and ranges from 0 (lowest rank) to 1 (highest rank). Personality was indexed by boldness, estimated by presenting individuals with a novel food (2 cm$^2$ pieces of potato or sweet potato dyed blue) while foraging naturally alone and quantifying the time that the individuals spent investigating—handling and smelling—the novel food (for further details, see *Carter et al., 2012b*). Individuals who investigated the novel food for longer were considered bolder. We tested individuals' boldness only once during the study period, but have previously found this test to be repeatable over three years (*Carter et al., 2012b*) and correlated with subjective ratings of boldness (*Carter et al., 2012a*). Age (in years) was estimated from a combination of known birth dates and dental patterns of tooth eruption and wear (see below). Unmarked immigrant males' ages were estimated at 9 years old when they appeared in the study troops, as this is the age most males were observed to transfer from our study groups.

## History of the study population

The Tsaobis baboons are a wild population that has been under study every austral winter since 2000. The field site is on private land, and the baboons have minimal contact with people other than the research team. The troops forage entirely naturally, except during specific research events that involve troop capture or feeding experiments. These occur very rarely (five occasions over the past 10 years), are short in duration (2–4 weeks), and entail the provisioning of the entire troop with corn kernels at a single site at dawn (e.g., *King et al. 2008*; *Carter et al., 2013*). Since 2009, individuals foraging alone have also been given the opportunity to sample a small, novel, food item (e.g., a slice of apple) at a random place and time, on average once per year, as a personality test (Carter et al., 2013).

During troop captures, all troop members are captured at dawn in individual cages baited with corn. They are sequentially anaesthetised using tiletamine–zolazepam and the entire troop is processed within a day, to be released together the following morning when fully awake. While the baboons are anaesthetised, age is estimated through dentition. Tooth eruption schedules are used to assign age up to molar eruption (*Kahumbu and Eley, 1991*), while age beyond this point is estimated from molar wear. Validation of this approach using individuals captured on multiple occasions (N = 19 over periods of 1–5 years) to compare estimated versus known age differences between captures indicates these estimates are robust (the mean difference between the observed and estimated time periods does not differ from zero: one-sample t-test, p>0.05; G. Cowlishaw, unpublished data).

## Quantifying the social networks

Individual associations were quantified using three proximity measures and two interaction measures.

### Proximity networks

In the first case, proximity data were collected from scans of individuals between dawn and dusk during all behavioural states. We recorded subgroups (hereafter 'groups') within each troop according to three proximity-based definitions of group membership. To ensure that groups were sampled

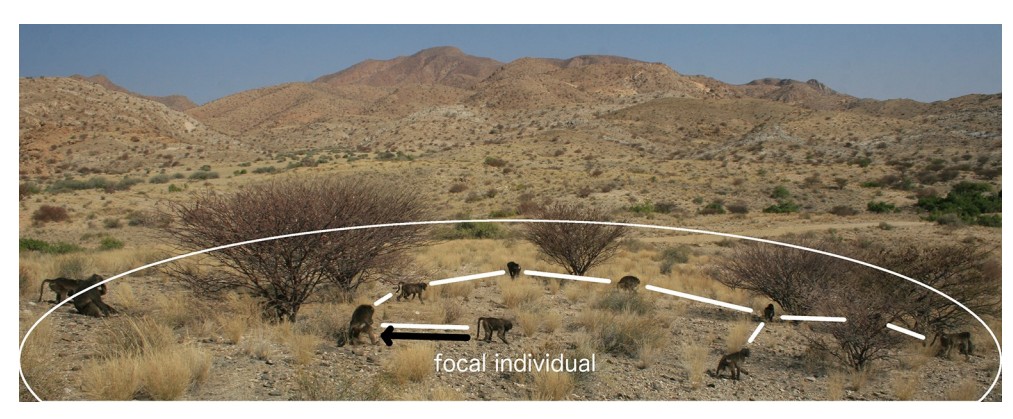

**Figure 1.** A visual representation of proximity methods used to define a connection. The black arrow represents a connection via the 5 m nearest neighbour rule; the white lines, connections via the 5 m chain rule; and the white circle represents the 10 m threshold distance for a connection (measures not to scale).

randomly and individuals were sampled evenly (and there was no bias against less social individuals), we quantified the groups associated with given 'focal' individuals chosen randomly from the troop membership. Because the troops can spread over 1 km² while foraging and finding particular individuals can be time consuming, the observer (MTT) searched for one of the first five individuals on the randomised list of baboons to optimise the number of independent groups sampled each day. Once a focal individual was found and its group membership quantified, that individual was removed from the list until all remaining individuals had been found and a new randomised list started. If a focal baboon had already been recorded in an already-sampled group (i.e., group membership had not changed), that individual was not sampled for an hour to ensure that the sampled groups constituted independent data. We are confident that each sampled group was independent and as such have not pooled data within arbitrary time periods (c.f. *Carter et al., 2009*). We recorded group composition at each scan for each of our three proximity rules: (1) the identity of all individuals within a 10 m radius of the focal individual (10 m scans), (2) the identity of all individuals whose most peripheral member was within 5 m of another individual of the group (5 m chain scan) and (3) the nearest neighbour within 5 m of the focal individual (nearest neighbour scans) (*Figure 1*) (*Castles et al., 2014*). Individuals who did not have a neighbour within the given distance for each proximity rule were recorded as alone.

## Interaction networks

Interaction data were recorded *ad libitum* by observers between dawn and dusk across all individuals as they moved continuously through the troop. On any given day, 1–4 observers were present with each troop, from a total pool of 7 observers for the field season. Observers collected *ad libitum* data while performing other data collection tasks at the site that required them to search for every individual every day (to perform the daily census, to perform the scans [this study] and while doing focal follow observations [not this study]) and to estimate group spread and activity every 30 min. Given that the observers were required to move constantly throughout the troop to monitor all individuals, our *ad libitum* data collection is not biased to more spatially central and/or obvious individuals and as such we have made no correction for individual baboon observability. New observers received training from experienced observers (who had worked with the baboons over at least two prior field seasons) until they could unambiguously and correctly identify dominance interactions (grooming was never ambiguous). No formal inter-observer reliability tests were done for *ad libitum* data, but we have successfully validated our training system for new observers collecting focal data in previous studies. If the interaction was ambiguous (as can sometimes occur during coalition formation), the data were not recorded.

We recorded both grooming and dominance interactions, noting the direction of the interaction. To quantify the dominance network, we used the same data as that collected to determine

dominance ranks. This is different to and independent of the analysis of the dominance ranks because it incorporates the frequency of interactions (in the case of weighted networks) as well as their direction (in the case of directed networks) (see below). To avoid pseudoreplication in the collection of the interaction data, an independent grooming bout was recorded when the partner identities of the dyad changed or the dyad stopped grooming and moved to a different location. As such, we did not record reversals of dyads within bouts (i.e., if individual A groomed B, and B then groomed A without moving to a new location, B was not recorded as grooming A). Furthermore, as sequential dominance interactions were pooled as one interaction (see above), only independent dominance events form these networks.

In total, we collected 6657 proximity scans including 2220 10 m scans ($N_J$ = 1091, $N_L$ = 1129, median scans per individual = 24, range = 13–24 scans), 2214 5 m chain scans ($N_J$ = 1085, $N_L$ = 1129, median = 24, range = 13–24), and 2223 nearest neighbour scans ($N_J$ = 1089, $N_L$ = 1134, median = 24, range = 13–24). We collected 23–24 scans for each individual for each proximity rule except for three individuals who were not present for the entire field season due to death or immigration ($N$ = ~13, 14, 17 scans each), but that were present during most of the patch experiments. We recorded 2768 grooming interactions in total ($N_J$ = 1331, $N_L$ = 1437; median per initiator = 16.0, range = 1–111). Finally, we recorded 1474 dominance interactions in total ($N_J$ = 618, $N_L$ = 856; median per initiator = 8.5, range = 1–116).

Social networks are made up of nodes (individuals) and edges (connections between the nodes). Network edges can have both weight and directionality. Weighted data, which we use in all our networks, indicate that the frequency of interactions between individuals is recorded, rather than a binary indicator of whether or not two individuals ever interacted. Directionality data indicate the degree to which interactions between individuals are reciprocal. Thus, undirected edges assume reciprocality, i.e. the relationships between dyads are equal and the association matrix is symmetrical along its diagonal, while directed edges assume non-reciprocality, e.g. A may groom B more than B grooms A and the association matrix is not symmetrical. From the records of group membership for the 10 m and 5 m chain rules, we created an undirected association matrix for each troop for each method using the simple ratio index (SRI): $x/(x + y_{ab} + y_a + y_b)$ where x is the number of times individuals A and B have been observed in the same group, $y_{ab}$ is the number of times individuals A and B have been observed in separate groups, $y_a$ is the number of times A has been observed without B and $y_b$ is the number of times B has been observed without A (*Cairns and Schwager, 1987*). The nearest neighbour data, like the interaction data, are directional (A may have B as a nearest neighbour, but B's nearest neighbour may not be A). Thus, for these datasets, we created directed association matrices which included the frequency with which A was the 'actor' towards B. For nearest neighbour, these frequencies represented the count of times that A had B as its nearest neighbour; for grooming, the count of times A groomed B; for dominance, the count of times A was aggressive towards B. However, dyads in the directed networks may acquire social information from each other equally. As such, we also analysed all our directed networks as undirected networks (by summing the interactions given and received by a dyad so that the association matrix was symmetrical along the diagonal). In total, across the 5 proximity and interaction datasets, this resulted in 5 symmetric and 3 non-symmetric (directional) weighted association matrices for each troop, resulting in a total of 16 association matrices (see *Figure 2* for diagrams of the networks according to the 5 association rules).

For two of the social networks (see below), we calculated two individual-level measures of network centrality hypothesised to be important for socially transmitted information (*Croft et al., 2008*): degree strength and betweenness. Degree strength (hereafter strength) sums the weighted edges each individual has with all other individuals in the network. Higher values indicate individuals who have more and/or stronger connections to others and are predicted to have greater access to information accordingly. Betweenness calculates the (weighted) number of shortest paths that travel 'through' a particular individual. Individuals with high betweenness connect different parts of networks and are therefore predicted to have greater access to information. We calculated both measures of centrality for the proximity, grooming and dominance networks using the igraph package (*Csardi and Nepusz, 2006*) in R (*Team, 2011*).

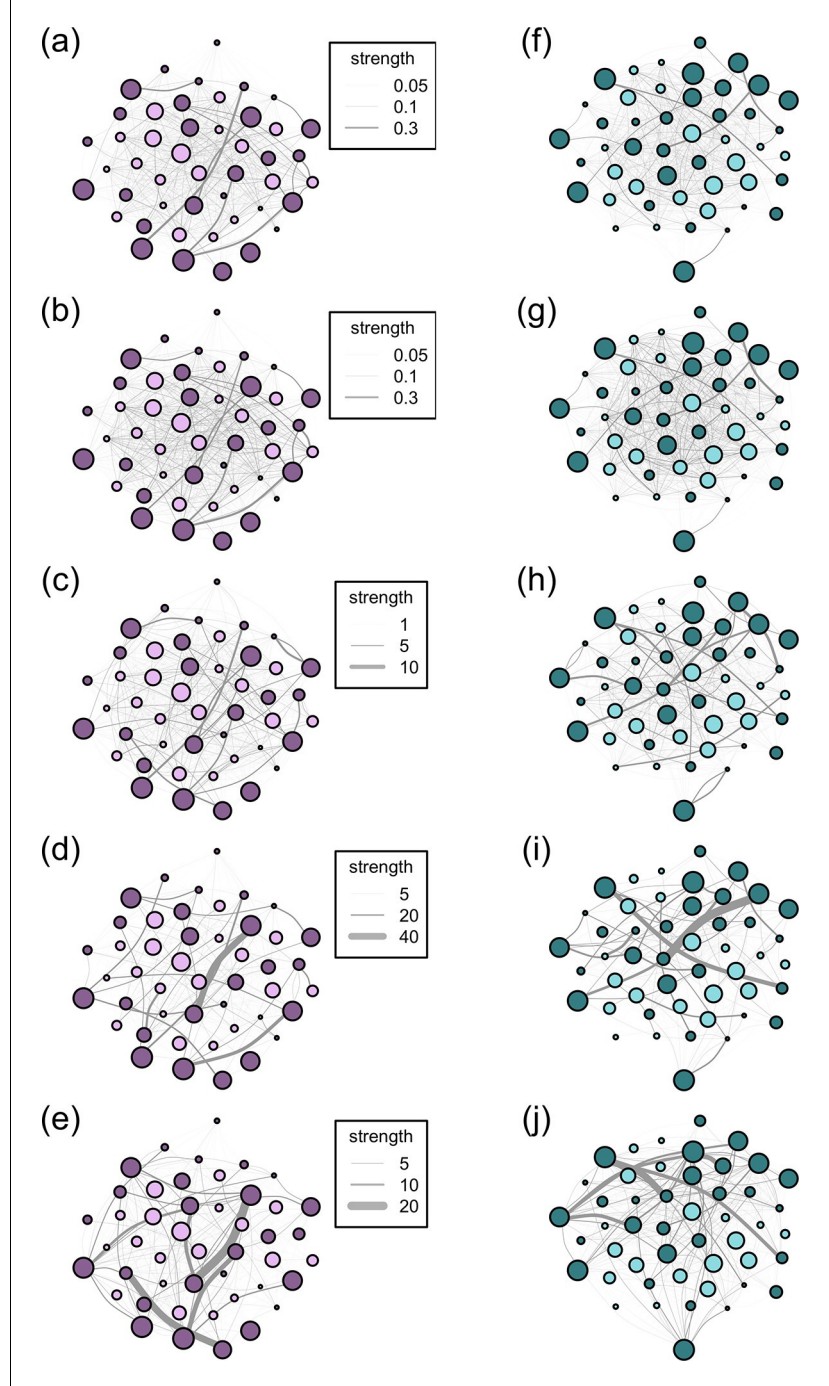

**Figure 2.** Networks diagrams created from the 5 association rules in two troops of baboons. Nodes (J troop: purple nodes, panels **a-e**; L troop: green nodes, panels **f-j**) represent individual baboons and edges between them indicate the strength of the measured relationship (see key). Presented are the networks based on the 10 m rule (**a**, **f**), 5 m chain rule (**b**, **g**), directed nearest neighbour rule (**c**, **h**), directed grooming interactions (**d**, **i**) and directed dominance interactions (**e**, **j**). Adults are represented by darker nodes, juveniles by lighter nodes (though we note that age was analysed as a continuous variable). Node size represents individuals' ranks, where larger nodes are higher ranks. Node positions are conserved between network diagrams in each troop.

The following figure supplement is available for figure 2:

**Figure supplement 1.** The relationships between social network metrics (strength and betweenness) within and between social networks created with five different rules for defining a connection between individuals.

## Information diffusion experiments

We experimentally assessed the diffusion of information about the location of newly discovered food resources by introducing patches of a highly preferred food, maize kernels, to the baboons. Two considerations were key to the design of the patch presentations: first, that the presentations were representative of naturalistic diffusions of information, such as about the location of a nest of eggs, and second, that the baboons did not learn to associate the observers with food. As such, one observer (AJC) created food patches by moving ahead of a foraging troop and scattering 52.9 ± 5.3 g of maize kernels over a 0.5 m$^2$ core area (with a little surrounding scatter enlarging this area to no more than 1 m$^2$), in the direction of travel of the troop. To avoid the baboons observing the patch being created, the observer either (i) quickly scattered the kernels as she was walking or (ii) pretended to get something from her field backpack while scattering the kernels behind her bag. In all cases, the baboons did not see the kernels being placed. Furthermore, because the observer was present in the troops for many hours preceding and following these trials, the baboons did not associate the camera nor the waiting behaviour of the observer with the presence of the patches. Because the foraging paths of the baboons are unpredictable, and the baboons typically have to be within 2 m of the patch to see the corn kernels (median, range 0–8 m, N = 38 trials with recorded detection distances), there was variation in the spatial position of the individual who discovered the patch. In 28 of the 50 experiments (56%), it was an individual at the leading edge of the group that found the patch. In 11 experiments (22%), it was an individual at the side periphery and in a further 11 (22%) an individual in the middle-back of the troop. In total, 37 different baboons (J = 19, L = 18) discovered the patches (median = 1 time, min = 1, max = 5). Not every patch that was put out was found by a baboon, either because passing individuals failed to detect it or the troop changed their direction of travel (N < 10). In such cases, the patches were picked up by the observer after the baboons had left the area, and excluded from the analysis. In total, we performed 50 successful information diffusion experiments (25 per troop).

One or two observers (which always included AJC) initially stood 15 m away from the patch and recorded each experiment using a video camera trained on the patch and surrounding area to dictate the identity and behaviour of any individuals coming within 25 m of the patch. The observer moved as required during the experiment to identify baboons. We recorded the identities of all individuals who (i) gazed at an individual in the patch, (ii) entered the patch and (iii) ate food from the patch (see *Videos 1* and *2*). These data were used to quantify social information (i) acquisition, (ii) application and (iii) exploitation, respectively. Note that the prolonged gazes observed during information acquisition were clearly distinct from the brief glances used in the routine monitoring of conspecifics, with most animals approaching and halting at the patch to watch its occupants (e.g., *Videos 1–3*). The experiment was conducted 25 times per troop. We subsequently counted the number of times each individual was recorded performing (i) – (iii) above, excluding those cases when they discovered the patch. On two occasions (in 50 experiments) an individual who acquired social information could not be identified before they left the vicinity of the patch, but otherwise we were able to identify all individuals who acquired, applied and exploited social information.

## Statistical analyses

### Identifying the diffusion path of information

To identify the network that best predicted information transmission, we conducted Order of Acquisition Diffusion Analyses (*Hoppitt et al., 2010*). OADA models information transmission

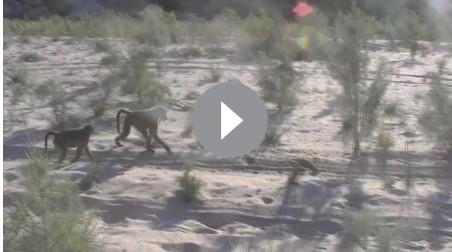

**Video 1.** Information diffusion experiments. The video shows rapid diffusion of information about the location of the food patch after its initial discovery. In the first experiment, several individual baboons successively enter the patch and are supplanted by more dominant individuals. In the second experiment, the patch is discovered by a low ranking female and then monopolised by high ranking juvenile male. The diffusion path is comparatively short. Please note that the videos were used to facilitate data extraction by dictating identities during rapid diffusions; we did not aim to capture all activity in the field of view of the camera.

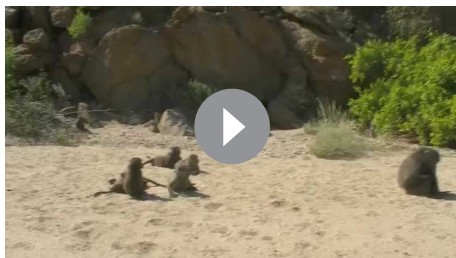

**Video 2.** Information acquisition, application and exploitation. The video shows an adult male monopolising the food patch, surrounded by juveniles who are obviously aware of the location of the patch, but cannot enter because of their lower rank. After the patch is depleted, the adult male exits the patch and many of the individuals subsequently apply the information they have acquired, even though it is outdated. One juvenile female (just off the bottom of the screen), has the lowest rank in the troop and could not apply the social information she had acquired. In this case, there was no social information exploitation, because the patch discoverer (the adult male) depleted the patch.

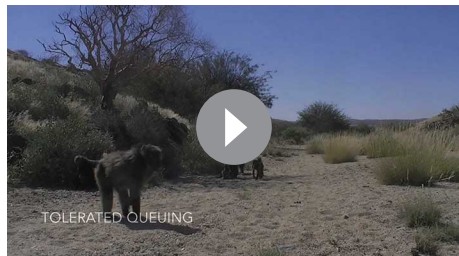

**Video 3.** Tolerated queuing, co-feeding and vocal protest. The first video (tolerated queuing) shows an adult male monopolising the food patch, with adult females and juvenile males and females queuing to check the patch after the male leaves. They enter the patch in the same order in which they were queuing. Two lower ranking adult females leave the area after queuing without entering the patch, demonstrating these females' unwillingness to apply the social information they had acquired after patch depletion. The second video (tolerated co-feed followed by protest) shows the initial patch discovery by an adult female who has come within 1 m of the patch (directly after she has been startled by the higher-ranking juvenile male foraging behind her). The pair subsequently co-feed in the patch, before the juvenile male vocally protests with pant-grunting.

by social connections versus asocial learning and fits the model estimates to the observed data (*Franz and Nunn, 2009*; *Hoppitt et al., 2010*). In the social transmission model, the rate at which naïve individuals acquire information from informed individuals is proportional to the connection(s) they have to those individuals. In the asocial model, information is acquired independently of the social network. The parameter $s$ estimates the social transmission relative to asocial transmission and ranges from 0, when there is no social transmission of information, to 1, when all information is transmitted socially (*Hoppitt et al., 2010*). In natural situations, $s \neq 1$ because at least one individual must acquire personal information for it to be socially transmitted. The social and asocial models are fitted to the observed order of diffusion data using maximum likelihood, and the model with the highest support, by comparison of Akaike Information Criteria corrected for small sample size ($AIC_C$) (or a likelihood ratio test), indicates the most likely route of information diffusion.

Previous studies applying network-based diffusion analysis to food patch discovery in social foragers have allowed for an 'untransmitted social effect' to accommodate the possibility that order of entry into a patch might not only reflect the pattern of social learning among associates but also the order in which the associates personally discover the patch (*Atton et al., 2012*; *Webster et al., 2013*). No such allowance was necessary here, because the transmission of social information about patch location was assessed directly through the visual monitoring of conspecifics in the patch rather than through the order of patch entry.

We fitted OADA models for the 50 diffusion events specifying each diffusion experiment with a task identity, each troop as a group and ties for individuals who acquired information simultaneously. We compared both additive and multiplicative OADA incorporating individual-level variables to control for possible sources of individual variation in asocial learning ability. We performed eight OADA models with social transmission (one for each of our eight networks) and one without social transmission. All nine models included rank, boldness, age and sex as individual-level variables and to determine the best model, we compared $AIC_C$s. After determining the network that best predicted the transmission of information (the 10 m network with multiplicative effects of the individual-level variables, see Results), we assessed which individual-level variables contributed to asocial learning by comparing the $AIC_C$s of models with all combinations of all individual-level variables, following *Hoppitt and Laland (2013)*. We estimated the effect size of each individual-level variable using

model averaging of those models with a ΔAICc≤2 (all of which were multiplicative models), but we present the AIC$_C$s of all multiplicative and additive models for comparison.

## Identifying phenotypic constraints on social information use

To identify the phenotypic constraints on the social information use sequence, we investigated whether individuals' (i) acquisition, (ii) application and (iii) exploitation of social information were affected by their phenotypes. Phenotypes were quantified according to social traits (dominance rank, network centrality), behavioural traits (personality), and demographic traits (age, sex). Two different measures of network centrality were used, namely the individual strength scores for the 10 m proximity and directed grooming networks, generating six phenotypic predictors in total. Individual betweenness scores were not used because they were strongly correlated with their corresponding strengths in almost all cases (*Table 2*; *Figure 2—figure supplement 1*). We chose to use the strengths and betweennesses from the 10 m proximity and directed grooming networks, because these were the best proximity and interaction network predictors of information diffusion respectively (see below). The 10 m proximity and grooming strengths were only marginally correlated with each other (*r* = -0.29) and below the level of collinearity concern (*Dormann et al., 2013*). All other combinations of phenotypic variables were similarly below the level of collinearity concern (*Table 3*).

We ran three generalised linear mixed models (GLMMs) in the lme4 package (*Bates and Sarkar, 2007*) with a Poisson link with the count of social information (i) acquisition, (ii) application and (iii) exploitation as the responses and troop as a random effect. For each response, we started with a full model comprising all six phenotypic predictors, and used backwards elimination of non-significant terms until we obtained the minimal model. Dropped terms were added to the minimal models to check significance.

All data used in these analyses are available online (*Carter et al., 2015*).

## Results

All individuals in both troops acquired social information in at least one experiment (barring one individual who acquired personal information of one patch, but died in the last week of experiments). On average (median), 10 individuals obtained information about the location of the patches (range = 2–27 individuals) in each diffusion experiment (see *Video 4* for an example of information diffusion through a network).

## Identifying the diffusion path of information

We found widespread evidence for the social transmission of information about the location of food patches (*Table 4*). All proximity networks and grooming networks had strong support for predicting the diffusion of information between group members in comparison to the asocial transmission model (ΔAIC$_C$ for the social transmission model with the lowest AIC$_C$ versus the asocial model = 192.9). In all cases, the multiplicative models had a better fit than the additive models. The dominance networks, however, had little support in either case. The social network that best predicted the transmission of information was the 10 m network (AIC$_C$ = 3968.9), followed by the 5 m chain network (AIC$_C$ = 3993.4, ΔAIC$_C$ = 24.5). These were followed by the undirected nearest neighbour network (ΔAIC$_C$ = 82.1), which performed better than the directed neighbour network (ΔAIC$_C$ = 114.4). All proximity networks were better at predicting diffusion than the grooming networks (*Table 4*). For both grooming and dominance, there was minimal difference in the performance of the models between the directed and undirected networks (ΔAIC$_C$ grooming = 3.6, dominance = 0.2). The social transmission parameter of the best multiplicative model (10 m proximity, *s* = 0.999) suggests that, following patch discovery, all subsequent discoveries were via social information. This was confirmed when comparing all possible combinations of individual-level variables in the 10 m model as all four of the best candidate models (ΔAIC$_C$ = 0) were multiplicative and the best additive model was comparatively poor (ΔAIC$_C$ = 9.85) (*Supplementary file 1*). Model-averaged estimates of the individual-level variables calculated from the multiplicative OADA models indicated that rank (β = 0.26), sex (β = 0.15) and age (β = -0.01) affected information diffusion, while boldness did not (β = 0.00), such that more dominant, younger male baboons were more likely to learn asocially about patch locations (see *Table 5* for a list of the parameter estimates of the competing models).

**Table 2.** Results of Spearman rank correlations testing whether there is a correlation between strengths and betweennesses in social networks created with different proximity and interaction rules. Presented is the rule, test statistic (*S*), rho (ρ), and *p*-value.

| Rule | S | ρ | p |
|---|---|---|---|
| 5 m chain | 217907.6 | -0.57 | <0.001 |
| 10 m | 220939.0 | -0.59 | <0.001 |
| Nearest neighbour directed | 81586.8 | 0.41 | <0.001 |
| Nearest neighbour | 9218.6 | 0.43 | 0.003 |
| Groom directed | 98412.9 | 0.30 | 0.005 |
| Groom | 15346.0 | 0.05 | 0.72 |
| Dominance directed | 67342.1 | 0.51 | <0.001 |
| Dominance | 7950.4 | 0.51 | <0.001 |

## Identifying phenotypic constraints on social information use

We found that individuals' phenotypes limited the acquisition, application and exploitation of social information about the location of food patches (*Table 6*, *Figure 3*). Proximity strength was the only predictor of information acquisition: more central baboons acquired social information more frequently. Proximity strength also showed a similar pattern with information application, but not exploitation. Information application and exploitation were further limited by sex and grooming strength, with males and more central individuals in the grooming network more likely both to apply and exploit acquired information. In combination with sex, dominance rank played a further limiting role on individuals' exploitation of information, such that females of low rank were most limited in their exploitation of patches. Finally, individuals' behavioural phenotypes, i.e., boldness, also influenced information exploitation. Overall, individuals that were better connected in the 10 m network were more likely to acquire social information, but it was higher ranking males who were more likely to exploit this information.

As a result of these constraints on successive steps in the social information use sequence, individuals only acquired social information about the patches on average (median) 6 times, applied social information 2.5 times, and exploited social information once (across a total of 25 trials per group). However, because of phenotypic variation, there was a substantial range around these medians. Thus, while the average individual acquired and exploited social information on <25% and <5% of occasions, respectively, others were able to acquire and exploit information on >50% and >35% of occasions, or not at all, depending on their phenotype.

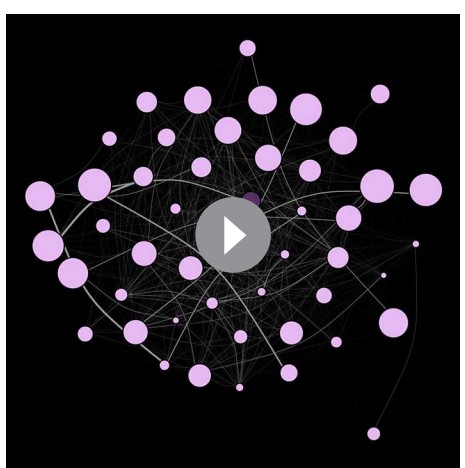

**Video 4.** Information diffusion through a social network. The animation shows the diffusion of information about the location of one of the food patches through the 5 m proximity social network of L troop. The nodes are scaled to the ranks of the individuals; the lines connecting the nodes are indicative of the strength of the connection between individuals. The nodes turn from pink to purple as they acquire social information about the location of the food patch by the initial discoverer (the original purple node).

## Discussion

Our study suggests that the route of information flow about ephemeral food patches is most closely matched by the 10 m proximity network, although the 5 m chain and nearest neighbour networks (both directed and undirected) also provided a good approximation. These results are consistent with several previous studies

**Table 3.** Correlation matrix of the phenotypes used in the analyses. Presented are the Spearman's rank correlation (S) estimates.

| Phenotype | Sex[a] | Age | Rank | Boldness | Proximity[b] | Groom[c] |
|---|---|---|---|---|---|---|
| Sex | 1 | | | | | |
| Age | -0.49 | 1 | | | | |
| Rank | 0.43 | 0.16 | 1 | | | |
| Boldness | 0.16 | -0.55 | -0.24 | 1 | | |
| Proximity | 0.29 | -0.60 | -0.02 | 0.51 | 1 | |
| Groom | -0.58 | 0.63 | 0.15 | -0.46 | -0.29 | 1 |

[a]coded as an integer: females = 0, male = 1.
[b], [c]Refer to strength in the identified network.

indicating the importance of spatiotemporal associations for information flow (e.g., great tits: *Aplin et al., 2012*; three-spine sticklebacks: *Atton et al., 2012*). While the most appropriate proximity network to capture information flow will vary depending on the type of information, the social forager, and the environment, our study emphasises the need to recognise that not all networks will be equally applicable. These findings build on the suggestions of previous authors (*Lehmann and Ross, 2011*; *Madden et al., 2011*; *Castles et al., 2014*), and provide the first quantitative demonstration that multiple networks should be considered when studying information transmission. In the present analysis, the grooming networks also provided a good match to the pattern of information flow, suggesting that individuals may be more likely to monitor those with whom they share strong social

**Table 4.** Comparisons of the additive and multiplicative OADA models with social transmission versus the asocial learning model.

| Model | Add/Multi | Predictor network | df | LogLik | AIC$_C$ |
|---|---|---|---|---|---|
| Social transmission | Add | 10 m | 5 | 1984.7 | 3979.4 |
| Social transmission | Multi | | 5 | 1979.4 | 3968.9 |
| Social transmission | Add | 5 m | 5 | 1992.9 | 3995.9 |
| Social transmission | Multi | | 5 | 1991.7 | 3993.4 |
| Social transmission | Add | NN directed | 5 | 2037.4 | 4085.0 |
| Social transmission | Multi | | 5 | 2036.6 | 4083.3 |
| Social transmission | Add | NN | 5 | 2024.4 | 4059.0 |
| Social transmission | Multi | | 5 | 2020.5 | 4051.0 |
| Social transmission | Add | Groom directed | 5 | 2045.7 | 4101.4 |
| Social transmission | Multi | | 5 | 2043.0 | 4096.1 |
| Social transmission | Add | Groom | 5 | 2045.2 | 4100.6 |
| Social transmission | Multi | | 5 | 2044.8 | 4099.7 |
| Social transmission | Add | Dom directed | 5 | 2076.8 | 4163.8 |
| Social transmission | Multi | | 5 | 2076.8 | 4163.8 |
| Social transmission | Add | Dom | 5 | 2076.6 | 4163.3 |
| Social transmission | Multi | | 5 | 2076.7 | 4163.6 |
| Asocial learning | - | - | 4 | 2076.8 | 4161.8 |

The predictor networks were the 10 m rule (10 m), 5 m chain rule (5 m), both of which were undirected, directed and undirected nearest neighbour rule (NN), directed and undirected grooming interactions (Groom) and directed and undirected dominance interactions (Dom). Presented are the models, degrees of freedom (df), -log-likelihoods (LogLik), corrected Akaike information criteria (AIC$_C$). Add/Multi refers to whether the model was additive (Add) or multiplicative (Multi).

**Table 5.** Parameter estimates of individual-level variables of the competing OADA models for asocial effects on social transmission in the 10 m networks.

| Model | Coefficient | Estimate | S.E. |
|---|---|---|---|
| 1 | Social transmission | 0.999 | |
| | Sex | 0.132 | 0.107 |
| | Rank | 0.498 | 0.174 |
| | Age | -0.024 | 0.012 |
| 2 | Social transmission | 0.999 | |
| | Boldness | 0.001 | 0.001 |
| | Age | -0.020 | 0.011 |
| 3 | Social transmission | 0.999 | |
| | Sex | 0.328 | 0.086 |
| 4 | Social transmission | 0.999 | |
| | Sex | 0.250 | 0.093 |
| | Rank | 0.353 | 0.158 |

Presented are the bounded social transmission estimates (for completeness), the fixed effects in the models and their standard errors (S.E.).

bonds, irrespective of the symmetry of those bonds (the model results for the undirected and directed networks were very similar). However, the absence of an effect of grooming strength on information acquisition at the individual level (**Table 6**) suggests that such effects are relatively weak in comparison to those of spatial proximity. In contrast, the dominance networks were entirely unrelated to information flow. Since information flow required visual observation, this suggests that the monitoring of conspecifics is independent of dominance rank. A recent study of social attention in wild vervet monkeys reported a similar pattern (**Renevey et al., 2013**). This may reflect the fact that dominants and subordinates monitor each other equally, albeit for different reasons: dominant animals seek to scrounge the food discoveries of others, while subordinates seek to avoid aggression.

In our analysis of phenotypic constraints on social information use, we identified three steps in the information use sequence: acquisition, application, and exploitation. In the first step, we found that information acquisition was independent of almost all phenotypic traits tested, i.e., age, sex,

**Table 6.** Parameter estimates of the minimal models investigating the effect of proximity and grooming strength on social information (i) acquisition, (ii) application and (iii) exploitation.

| Response | Predictor | Effect size | S.E. | t | P |
|---|---|---|---|---|---|
| Social information acquisition | Intercept | 0.23 | 0.21 | 1.11 | 0.27 |
| | Proximity strength | 0.66 | 0.07 | 8.87 | <0.001 |
| Social information application | Intercept | -1.30 | 0.38 | -3.40 | <0.001 |
| | Proximity strength | 0.65 | 0.10 | 6.45 | <0.001 |
| | Grooming strength | 0.01 | <0.001 | 4.74 | <0.001 |
| | Sex[a] | 0.84 | 0.15 | 5.48 | <0.001 |
| Social information exploitation | Intercept | -2.43 | 0.39 | -6.17 | <0.001 |
| | Grooming strength | 0.02 | <0.001 | 4.67 | <0.001 |
| | Sex[a] | 0.73 | 0.33 | 2.21 | 0.03 |
| | Boldness | 0.01 | <0.001 | 3.74 | <0.001 |
| | Rank | 1.39 | 0.51 | 2.71 | 0.01 |

Presented are the predictor variables, their effect sizes, standard errors (S.E.), t values and p-values.

[a]Reference category: female

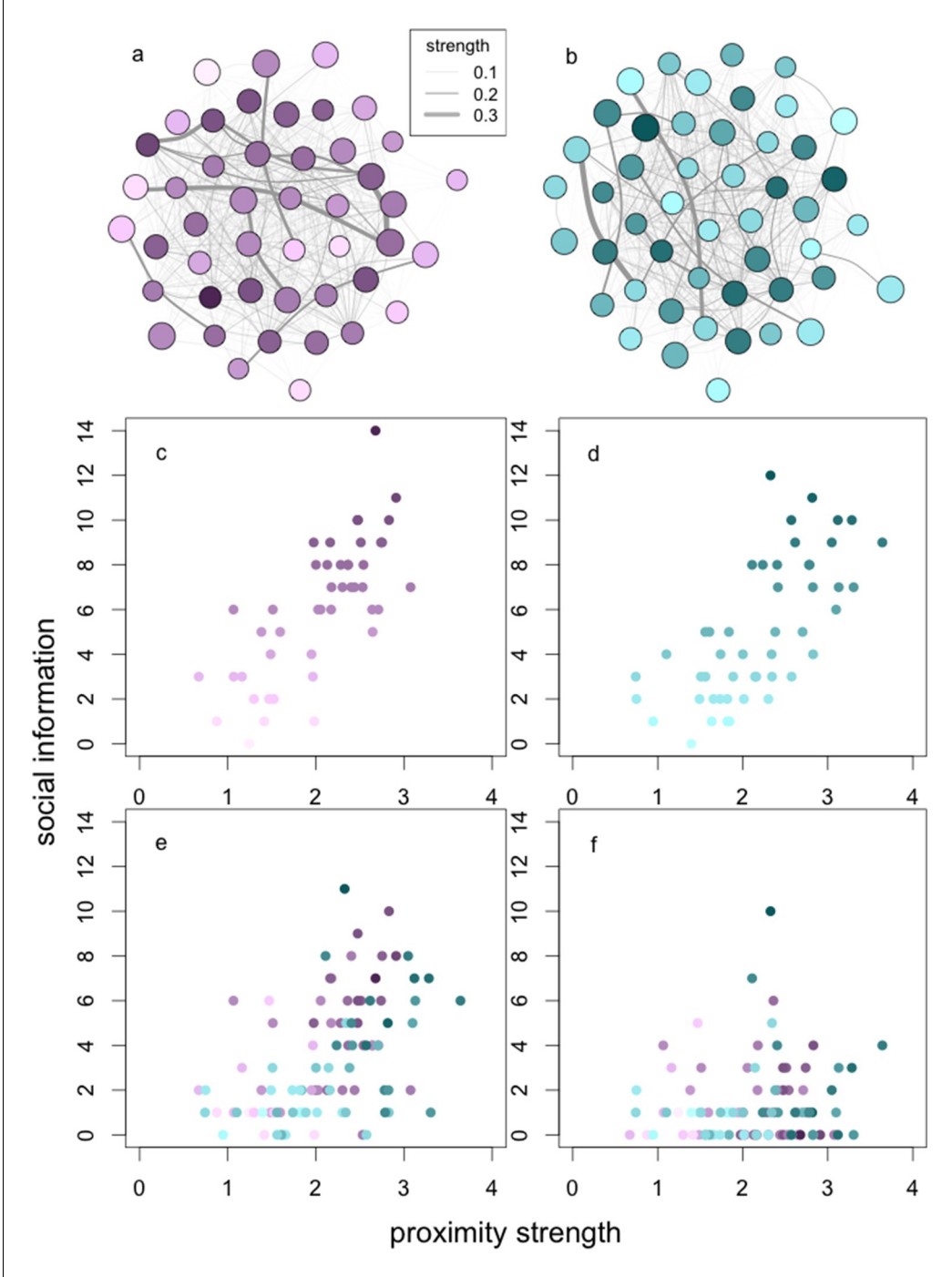

**Figure 3.** The relationships between social network centrality and successive steps of the social information process. The relationships between social information (**c, d**) acquired, (**e**) applied and (**f**) exploited by wild baboons and their degree strengths in the social networks. Presented are the proximity networks from which degree strengths were calculated for (**a**) J and (**b**) L troops, where nodes represent individuals, node size represents the rank of the individual, and node luminance represents the number of times the individual acquired information (darker nodes acquired social information on more occasions; this colouration is conserved throughout the figure). Lines connecting nodes represent the strengths of the connections between dyads where thicker lines are stronger connections (see legend). Presented below the networks is the relationship between (**c, d**) social information acquired in (**c**) J and (**d**) L troops, (**e**) social information applied and (**f**) social information exploited (both troops plotted together).

rank, personality, and social bonds (grooming strength). The only important trait was individual centrality in the 10 m proximity network. This suggests that visual information about the patch was inexpensive to collect, and limited only by an individual's spatial associations with other group members. Where the acquisition of social information is more costly, we might expect other phenotypic traits to become important. For instance, among juvenile chimpanzees (*Pan troglodytes*), sex differences in attentiveness are believed to explain why females spend more time observing, and are faster to learn, the challenging skill of 'termite fishing' (*Lonsdorf, 2005*).

In the second and third steps of the information use sequence, the application and exploitation of information were closely linked. Both steps involved patch entry, but the latter also involved the successful capture of foraging benefits from the patch. The similarities and dissimilarities in phenotypic predictors between the two models are highly informative, not only with respect to understanding the different constraints that can operate along the information use sequence, but also in elucidating how information use mediates the acquisition of monopolisable resources. The latter is possible because our experimental design makes information exploitation synonymous with resource acquisition. Clearly, in many other cases, information exploitation will be unrelated to monopolisable resources but rather involve other types of knowledge, such as foraging skills, predation risk, and mate compatibility. In these instances, the phenotypic constraints on information application and exploitation may be quite different to those observed here.

We begin our assessment of phenotypic constraints on information application and exploitation with dominance rank. Dominant animals were far more likely to successfully exploit social information, because they were able to monopolise food patches. Indeed, as information about the patch spread, increasingly dominant animals would become informed and enter the patch, supplanting lower ranked occupants and preventing others from entering subsequently until the patch was exhausted. This pattern is consistent with how dominant animals scrounge from others in this population (*King et al., 2009*; *Marshall et al., 2012*) and across social foragers generally (*Barta and Giraldeau, 1998*). Surprisingly, however, dominance did not predict information application. The reason for this is that many subordinates also entered the patch, but only after the dominant animal had left (see *Video 2*). Some of these animals were late arrivals, but a large number would be waiting ('queuing') nearby for the dominant animal to leave. Since the patch was largely depleted when the dominant animal left, the subsequent patch entries by subordinates imply the application of outdated information, at a surprisingly large scale given over half of those entering the patch failed to exploit it. One possible explanation for such apparently maladaptive behaviour is that, while baboons are able to collect social information about patch location, they are unable to collect 'public' information about patch quality. A similar pattern has been observed in three-spined sticklebacks (*Coolen et al., 2003*). However, other experimental work in this population indicates that baboons are able to collect such information (*Lee, 2015*). A more likely explanation is that lower ranking individuals are aware that the patch is largely empty, but there is minimal cost in entering it and sifting through the sand in case any food items remain. Indeed, in a small number of cases, late arrivals did find a small number of kernels that had been overlooked.

Proximity strength remained a strong predictor for information application, as might be expected: individuals had to find the patch before they could enter it. However, because only a very small fraction of those who acquired and applied the information were able to benefit from it, the effect of proximity strength was lost in information exploitation. Grooming strength, however, was important for both information application and exploitation, possibly reflecting the role that social bonds play in tolerance at feeding sites, both in this baboon population (*Sick et al., 2014*) and in primates more generally (*Ventura et al., 2006*; *Tiddi et al., 2011*). Nevertheless, tolerance of direct co-feeding at the patch (patch sharing) was rare in this experiment: of the 293 patch entries observed in the 44 experiments for which these data could be extracted, only 14 (4.8%) resulted in co-feeding (where two or more individuals fed simultaneously from the patch). Moreover, in 4 of these cases (including one group of 3 co-feeders), there were vocal protests of intolerance from one or both parties (see *Video 3* for an example of a vocal protest during co-feeding). Thus toleration of co-feeding could be said to occur on only 9 occasions (3.1%). Instead, the tolerance observed in this experiment was primarily of close proximity of individuals to the patch, which allowed individuals to queue for and quickly enter the patch on a dominant's exit (see *Video 3*). There are thus two strategies for information application to occur, both of which may rely on social bonds: tolerated co-feeding and tolerated (close-proximity) queuing. A change in the dimensions of the patch to allow more

foragers concurrently to occupy it, and thus a greater possibility of tolerated co-feeding, may show a greater effect of grooming strength on both social information exploitation and application.

Sex also played a role: females were less likely to apply and exploit social information. As we control for rank in the analyses, this finding may reflect female reproductive constraints. Previous work on nine-spine sticklebacks (*Pungitius pungitius*) has indicated that female gravidity can influence the use of social information (*Webster and Laland, 2010*). In our case, many of the observed females were either pregnant or lactating, and females in these states experience higher foraging demands (*Silk, 1987*; *Barrett et al., 2006*), potentially making them less willing either to forego valuable foraging time to queue for patch entry (information application) or to spend excessive time searching for food once in the patch (information exploitation). Shy animals also showed lower rates of information exploitation, presumably because they were more nervous and therefore spent more time in the social monitoring of conspecifics (*Edwards et al., 2013*). The observation that bolder animals were more likely to successfully exploit information is consistent with the finding that bolder animals are also more likely to demonstrate social learning in this population (*Carter et al., 2014*). In comparison, *Harcourt et al. (2010)* found no effect of boldness on social information use in three-spined sticklebacks. However, that study only went as far as the information application step of the information use sequence. We only found an effect of personality in the final information exploitation step.

Phenotypic variation was also observed in asocial learning. Younger, more dominant males were more efficient at finding food patches. The most likely explanation for this pattern is that dominant juveniles were more likely to be at the leading edge of a foraging group, and therefore the first to encounter the patches. Similarly, juvenile ring-tailed coatis (*Nasua nasua*) occupy positions at the leading edge of their foraging groups (*Hirsch, 2011*). The multiplicative effect further suggests that the probability of younger dominant males occupying this spatial position increased the better connected they were in their social network. Notably, we found no effect of age on social information use. While this is not surprising at the information acquisition stage, where there were no phenotypic constraints other than network position, it is more surprising for information application and exploitation, especially since juveniles appear to show greater propensity for social learning than adults, both in baboons (*Carter et al., 2014*) and more generally (e.g., meerkats, *Suricata suricatta*: *Thornton and Malapert, 2009*). However, this propensity usually refers to social learning tasks involving novelty or complexity, reflecting the tendency for juveniles to be more curious and exploratory about their environment (*Reader and Laland, 2001*; *Kendal et al., 2005*; *Benson-Amram and Holekamp, 2012*). In our case, the task simply involved entering and exploiting a patch of familiar food, and for such a task it might be expected that individuals of all ages would show equal abilities.

Our study raises a further point about the type of social information that is transmitted. We provided individuals with the opportunity to acquire social information about the location of a highly preferred food source in a novel location. These novel patches were rapidly depleted and the social information quickly became outdated. Such short-lived, ephemeral information may be easy and cheap to acquire and, as we have found, more likely be transmitted through proximity rather than interaction networks. Whilst our results support previous findings that proximity social networks are important for the transmission of ephemeral and/or easy-to-acquire information among individuals in the wild (*Aplin et al., 2012*), evidence from starlings suggests that more complex information diffuses through alternative networks. *Boogert et al. (2014)* showed that information about novel foraging skills diffused through the perching rather than foraging social networks. Whilst it is challenging to determine the difficulty with which species acquire different types of information (*Griffin et al., 2015*), research is needed to elucidate how the type of social information influences how quickly and through which networks it is transmitted. In the baboon system, we might similarly expect that social information that takes longer to acquire and/or process will transmit through different networks to those that transmit easy-to-acquire information.

Our finding that phenotype limits information use builds on previous work indicating that individual state, such as uncertainty or the possession of outdated information, can influence social learning strategies (reviewed in *Rendell et al., 2011*). Our study extends this work, revealing fundamental individual differences in the ability to use social information. Decomposing social information use into three constituent steps has further illuminated how these individual differences limit information use. The implications of such sequential phenotypic constraints are manifold. We conclude by considering two points. First, only a small number of individuals who acquire or apply social information may successfully exploit it. Similarly, *Racine et al. (2012)* report that ring-billed gulls (*Larus*

*delawarensis*) may acquire social information about the location of food resources but be unable to apply it because of parenting constraints. Second, where phenotypic traits are expressed in relation to conspecifics (e.g., dominance rank, network position), social information use will vary markedly between social environments. For instance, in a captive setting where competition is minimised (e.g. by testing individuals when isolated and not at risk of aggression: *Call et al., 2005*) (c.f. *Drea and Wallen, 1999*), subordinate animals may show a propensity for copying others (e.g. *Kendal et al., 2015*), but in the wild, where subordinates are more vulnerable to aggression and where food resources are relatively more valuable, subordinates may rarely copy others. Together, these two points highlight a critical disjunction between an ability to acquire information and to capture its benefits. This disconnection is likely to have a fundamental impact on selection for social information use, such that even in social information-rich environments, only a small number of individuals of a particular phenotype may be selected to use it.

## Acknowledgements

We give a big Ooh-la-la shout out to Alice Baniel, who generously allowed us to parasitise her field season and organise ours within hers. We thank the members of the Tsaobis Baboon Project 2014 Team 1 for collecting many of the interaction data and Neeltje Boogert for putting up with endless questions about interpretations of model parameters in OADA. We thank Alex Lee for being patient with the installation of MatMan and calculating the dominance hierarchy. We thank Will Hoppitt for answering our questions about model averaging in OADA. We thank Brianne Beisner, Brenda McCowan and an anonymous reviewer for constructive feedback on the manuscript. AJC is supported by a Junior Research Fellowship from Churchill College, which also supported the costs of this field work. We are grateful to the Ministry of Lands and Resettlement for permission to work at Tsaobis Leopard Park, the Gobabeb Training and Research Centre for affiliation, and the Ministry of Environment and Tourism for research permission in Namibia. We are also grateful to the Snyman and Wittreich families for permission to work on neighbouring farms. This paper is a publication of the ZSL Institute of Zoology's *Tsaobis Baboon Project*.

## Additional information

### Competing interests
AJC: Early Career Advisor for *eLife.* The other authors declare that no competing interests exist.

### Funding
No external funding was received for this work.

### Author contributions
AJC, Conception and design, Acquisition of data, Analysis and interpretation of data, Drafting or revising the article; MTT, Acquisition of data, Drafting or revising the article; GC, Conception and design, Analysis and interpretation of data, Drafting or revising the article

### Author ORCIDs
Alecia J Carter, http://orcid.org/0000-0001-5550-9312

### Ethics
Animal experimentation: Our study was conducted using protocols assessed and approved by the Ethics Committee of the Zoological Society of London, and approved by the Ministry of Environment and Tourism in Namibia (Research/Collecting permit 1892/2014).

## Additional files

### Supplementary files

• Supplementary file 1. Comparisons of the different models used to assess the effect of individual-level variables on the transmission of information among individuals, with ΔAICc <2 considered to have good support. Shown are the combinations of individual-level variables, whether the model was multiplicative, additive or neither (when the individual-level variables were not included) (model type), whether individual-level variables were included (0 = no, 1 = yes); whether the process was modelled as a social or asocial diffusion (social/asocial); the AICc of the model; the difference from the AICc of the best model (ΔAICc); the support for the model (model weight); and the relative weight of the model in comparison to the model set.

### Major datasets

The following dataset was generated:

| Author(s) | Year | Dataset title | Dataset URL | Database, license, and accessibility information |
|---|---|---|---|---|
| Carter AJ, Torrents Ticó M, Cowlishaw G | 2015 | Data from: Sequential phenotypic constraints on social information use | https://dx.doi.org/10.6084/m9.figshare.2010396 | Available at figshare under a CCBY Attribution licence |

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
