## [Decision Letter]

Thank you for submitting your work entitled "Sequential phenotypic constraints on social information use" for consideration by *eLife*. Your article has been favorably evaluated by three reviewers, including Brianne Beisner. Brenda McCowan and a guest Reviewing Editor, and the evaluation was overseen by Ian Baldwin as Senior Editor.

The reviewers have discussed the reviews with one another and the Reviewing Editor has drafted this decision to help you prepare a revised submission.

Summary:

This study takes a novel approach to the analysis of information transmission and decision-making by viewing it as a three-step process of information acquisition, application and exploitation. Using social network analysis and diffusion analysis, it identifies the type of social network that best describes the diffusion of information about the location of small, novel artificial food patches for two troops of wild chacma baboons. It subsequently examines the role of several characteristics that govern the decision-making process at each of the three steps. The application of this novel approach is an important contribution that could have a significant, innovative influence on future research into decision-making and social transmission of information. The analysis presented in this paper goes beyond many other papers that use social network approaches by investigating multiple different social networks, which is critical to gaining a full understanding of social information transmission, because social groups are comprised of many different social networks. The data analysis techniques appear to be appropriate and state of the art. The article is well-organized and well-written. Although some of the results presented are not surprising and seem very intuitive (e.g., dominant animals were better able to exploit the information acquired about the food patches), other findings are quite valuable (e.g., boldness was relevant only at the stage of successful exploitation; proximity networks best predict social information transmission whereas dominance networks did not at all predict transmission). Our major concerns are primarily about details about the methods used to carry out the field experiments.

Essential revisions:

1) More information would be useful about the habitat, terrain, diet and foraging habits of the baboons at the study site. Although the authors say that individual baboons had to be within 1 m of the patch to see the maize, it would be helpful to understand how the authors determined this distance and whether this distance was the case in all trials. The reader is unable to judge the degree to which baboons may be able to spot kernels of maize from a greater distance without this information. Indeed, in the videos, the maize seems to be placed in areas bare of vegetation where it is plausible to think that it could have been spotted from more than 1 m away. Also, the videos suggest that the maize may have been concentrated over a much smaller area than 1 square meter, making spotting easier, but co-feeding less likely to occur.

2) Please give a breakdown of the age/sex composition of each group, and indicate whether all individuals over 2 years of age were individually recognizable to all observers.

3) It would be useful to distinguish between exploitation of patches in two different ways-via co-feeding with another individual, or waiting until the patch is vacated. The authors mention the waiting technique and allude to issues of tolerance, but do not state explicitly how many entrances into the patch and how much exploitation of the patch involved the two different strategies. If the area over which the maize was spread varied or tended to be smaller than the 1 m cited (see above), this could have affected the likelihood of individuals pursuing each strategy, and hence the importance of such factors as grooming strength. We suggest that both possible types of exploitation be described early in the paper to prepare the reader for further distinction in the data later. Re-analysis taking the two strategies into consideration would be useful, if possible. If not, discussion of this issue is needed.

Clarifying this issue should also help in the interpretation of the role of boldness in exploitation. The prior test of boldness concerned handling of a novel food in isolation, i.e., boldness in a nonsocial context. The nonsocial test of taking a novel food item in isolation is probably assessing the animal's boldness with regard to exploration. Yet the application of boldness in the study likely had a social element to it, particularly when/if individuals attempted to co-feed with others in these small patches. Hence, the choice of the word 'shy' (Discussion, fifth paragraph) to mean the opposite of bold may not be accurate. What evidence is there that the two forms of boldness are equivalent? Correlated? Certainly humans differ in social vs. nonsocial manifestations of boldness. If they are not known to be equivalent, it would probably be more accurate to avoid the word 'boldness' and to say instead that baboons that show a high tendency to explore novel objects and surroundings also had a higher rate of successfully exploiting the experimental food patches.

4) How were dental patterns of tooth eruption and wear observed? Were the baboons trapped and sedated? A brief background section on the history, management and handling of the population would be useful. Were their diets supplemented by humans in the past? If so, how was it done?

5) There was some confusion among the reviewers about how the data on grooming and dominance interactions were collected. Certainly the authors took pains to collect proximity data in a systematic manner that allowed them to compare individuals. However, it is less clear that this was the case for grooming and dominance-related interaction. The authors state that "Interaction data were recorded ad libitum by all observers at the project (n=6-7 in total) between dawn and dusk across all individuals as they moved continuously through the troop". Given that troops could be dispersed over 1 square km, was what they did equivalent to all occurrences sampling? Could the entire troop be observed at once and all instances of grooming and agonism recorded? Do the authors mean that all 6 or 7 observers collected data at the same time, working different parts of the troop? If so, what steps were taken to make sure coverage was even and that individuals were not sampled more than once over a short interval of time? Was the amount of time each individual was observed recorded and used to calculate rates? Was any other sort of correction made for variation in the observability of individuals? If not, this could have seriously biased the social networks they derived. Given that the ranges of grooming interactions and dominance interactions per initiator were so large (1-111 and 1-116, respectively), we suggest that it may be better to cite the means and ranges of total grooming and dominance interactions per subject rather than initiations.

6) Please clarify how dominance networks are different from dominance ranks for readers that are less familiar with social network analysis. Please also discuss your use of undirected and directed networks. While OADA can be used with both directed and undirected networks, the directionality of the grooming and dominance networks (and lack of directionality of the proximity networks) needs to be discussed more fully. We suggest the authors explain (a) how directionality affects their interpretations of their results and (b) if undirected networks were attempted for grooming and aggression to see whether pairs that interact aggressively/ submissive frequently also appeared to have monitored each other frequently enough to pass on social information.

7) Six or seven observers were used. Were interobserver reliability tests done? If so, how and what was the criterion for passing?

8) Figure 1, please provide median and range for number of scans per individual.

9) The authors state that they hid the food when out of sight from the troops, but stood 15 m away from it with a video camera as the monkeys approached. What steps were taken to insure that the presence of observers at this point did not provide cues to the monkeys that a food patch was nearby? Cues of this nature could potentially pose a serious flaw in the design of the experiment, especially given that baboons catch on quickly to such things, and the experiments were replicated 25 times for each troop.

10) More than half the time the patch was found by the leading edge of the troop. Please discuss how this may have influenced the outcome of the study and the results indicating the importance of position in the proximity network. Are the most central animals disproportionately found in the leading edge? Would it be possible to control for the part of the group that happened to encounter the patch first-or run separate analysis for each condition? A brief description of the attributes of animals in different positions of an advancing group might help interpret the findings. How many times was the patch not found by the baboons? Did the 50 experiments include only those trials in which the patch was found?

11) The determination of social acquisition of information about the patch was via gazing of an individual toward another individual that was in a patch. Given that baboons constantly monitor each other's positions with gaze, were frequencies of gazing corrected for baseline rates of gazing at other individuals not in the patch? Was there a difference, e.g., in the duration of gazes, that would indicate the likelihood that information was actually acquired and that the gaze was not a routine monitoring of position only?

12) Results subsection “Identifying phenotypic constraints on social information use”, last paragraph and elsewhere. What does 'average (median)' mean? Average (meaning mean) and median are the same?

13) In the Discussion, the authors should explicitly consider the implications of using only small and ephemeral food patches. The brief nature of their food patch experiments means that animals that acquired information about the location of the food patch (and perhaps even applied it, though unsuccessfully) did not have the opportunity to apply and exploit such information in the future. From an evolutionary perspective, however, social information can spread about more permanent food patches as well as ephemeral food patches, and there may be future opportunities for animals to return to the location of more permanent food patches and they may be able to apply the information acquired (about this new food patch) as well as successfully exploit it. This may even be a reason for low-ranking individuals to enter a depleted food patch (especially if visibility of patch details is somewhat limited without entering it) – to determine for themselves whether the food patch may be a permanent patch that they can return to at a later date.

---

## [Author Response]

*1) More information would be useful about the habitat, terrain, diet and foraging habits of the baboons at the study site. Although the authors say that individual baboons had to be within 1 m of the patch to see the maize, it would be helpful to understand how the authors determined this distance and whether this distance was the case in all trials. The reader is unable to judge the degree to which baboons may be able to spot kernels of maize from a greater distance without this information. Indeed, in the videos, the maize seems to be placed in areas bare of vegetation where it is plausible to think that it could have been spotted from more than 1 m away. Also, the videos suggest that the maize may have been concentrated over a much smaller area than 1 square meter, making spotting easier, but co-feeding less likely to occur.*

We have added the following text to explain the field site in more detail (Materials and Methods, subsection “Study area and study species, first paragraph”): “Two habitat types make up the Tsaobis terrain: open desert and riparian woodland. […] The baboons’ main predator, the leopard (*Panthera pardus*), is rare at Tsaobis and the risk of predation is low.”

Regarding the distance from which the patch was noticeable, we returned to the videos to code estimated distances from which the patches were spotted. Depending on when we started recording (we only recorded on discovery of the patch to save video camera battery and memory card space), we were able to extract distances for 38 patch trials. The median distance was 2 m (range = 0-8). We note that the missing distances were not random – we were more likely to miss the beginning (seconds) of a trial if the patch was found when an individual was in close proximity to it and on several occasions patches were missed by baboons even when they were in close proximity to it – on two occasions individuals walked over the top of the patches and did not notice them! However, we have amended the manuscript to replace our previous estimate of 1 m with the calculated median and range values (Materials and Methods, subsection “Information diffusion experiments”, first paragraph).

The reviewers are also correct that the majority of the kernels were concentrated in a smaller area (0.5 m^2^), but in most cases several kernels were beyond this area and could spread up to 1 m across. This is because we did not want the baboons to associate the observer’s behaviour with creating a patch, and as such we created the patches as quickly as possible (see response to Query 9 below for more details on our strategy), which meant that not all of the kernels were within our small patch area. We have amended the text to reflect this more precisely (Materials and Methods, subsection “Information diffusion experiments”, first paragraph): “…one observer (AJC) created food patches by moving ahead of a foraging troop and scattering 52.9 ± 5.3 g of maize kernels over a 0.5 m^2^ core area (with a little surrounding scatter enlarging this area to no more than 1 m^2^), in the direction of travel of the troop.”

*2) Please give a breakdown of the age/sex composition of each group, and indicate whether all individuals over 2 years of age were individually recognizable to all observers.*

We have now provided this information (Materials and Methods, subsection “Study area and study species”, second paragraph).

*3) It would be useful to distinguish between exploitation of patches in two different ways-via co-feeding with another individual, or waiting until the patch is vacated. The authors mention the waiting technique and allude to issues of tolerance, but do not state explicitly how many entrances into the patch and how much exploitation of the patch involved the two different strategies. If the area over which the maize was spread varied or tended to be smaller than the 1 m cited (see above), this could have affected the likelihood of individuals pursuing each strategy, and hence the importance of such factors as grooming strength. We suggest that both possible types of exploitation be described early in the paper to prepare the reader for further distinction in the data later. Re-analysis taking the two strategies into consideration would be useful, if possible. If not, discussion of this issue is needed.*

This is an interesting idea. Following the reviewers’ suggestion, we now state explicitly in the Discussion how many instances of co-feeding there were (fifth paragraph) and summarise them for the reviewers here. We extracted instances of co-feeding from 44 experiments’ videos (due to a technical problem with one of the video cameras, we could not download 6 of them from the camera we used and consequently could not back them up). In the 44 trials, there were 14 instances of co-feeding i.e. individuals eating >1 kernel from the patch simultaneously (10 pairs, and 3 triplets); of these 14 co-feeding patch entries, 4 were not ‘tolerance’ per sei.e. two individuals fed at the patch but at least one of them vocally protested against the other. In total, there were 293 patch entries in this subset of 44 experiments; thus the vast majority of patch entries were not tolerated co-feeding. The tolerance that we observed was primarily for individuals to be allowed in close proximity to an occupier of a patch without being threatened or chased away. Tolerated queuing could thus be defined as a patch occupier allowing the proximity of an individual within, for example, 10 m of the patch for >10s. Unfortunately, as we did not set out to quantify this behaviour in these experiments, we cannot retrospectively accurately estimate instances of tolerated queuing – as our goal was to quantify the identities of individuals acquiring, applying and exploiting information, we did not film or keep an accurate dictated record of all those individuals who might be waiting around the patch, their distances from the patch, and their turnover. In our future experiments, we could endeavour to collect these data with the use of a second observer. As suggested by the reviewers, we discuss these two strategies in the text, suggest a change in the experimental design that could test this idea, and provide another video of “tolerated queuing” to illustrate this strategy in individuals of different ranks (as opposed to the infants queuing for the patch of the dominant male in Video 2) (Discussion, fifth paragraph).

*Clarifying this issue should also help in the interpretation of the role of boldness in exploitation. The prior test of boldness concerned handling of a novel food in isolation, i.e., boldness in a nonsocial context. The nonsocial test of taking a novel food item in isolation is probably assessing the animal's boldness with regard to exploration. Yet the application of boldness in the study likely had a social element to it, particularly when/if individuals attempted to co-feed with others in these small patches. Hence, the choice of the word 'shy' (Discussion, fifth paragraph) to mean the opposite of bold may not be accurate. What evidence is there that the two forms of boldness are equivalent? Correlated? Certainly humans differ in social vs. nonsocial manifestations of boldness. If they are not known to be equivalent, it would probably be more accurate to avoid the word 'boldness' and to say instead that baboons that show a high tendency to explore novel objects and surroundings also had a higher rate of successfully exploiting the experimental food patches.*

The reviewers have raised two of the perennial issues with personality studies: (1) how to label traits and (2) how to interpret cross-context correlations in behaviour. Regarding the first issue, we have shown convergent validity for this trait using a multimethod approach (Carter et al. 2012), which suggests that (i) this trait is more general than a “tendency to explore novel objects” and (ii) this trait is not exploration, though there may be overlap with such a trait (Carter et al. 2013). As such, we feel it is better to retain the term ‘boldness’. In addition, this approach is consistent with the terminology we have established in our past publications (while using the term “exploration” now would introduce confusion). Regarding the second issue, we have three lines of evidence that novel food boldness also relates to the social environment: novel food boldness predicts social learning (Carter et al. 2014), bolder individuals tend to associate with other bolder individuals (Carter et al. 2015), and novel food boldness correlates with cross-context observer assessment scores of boldness (Carter et al. 2012). Since one of the goals of personality research is to understand such overlaps (e.g. behavioural syndromes, Sih et al. 2004), we feel the current presentation is in-line with these approaches.

Carter, AJ; Marshall, HH; Lee, AEG, Torrents Ticó, M; Cowlishaw, G. 2015. Phenotypic assortment in wild primate networks: implications for the dissemination of information. Royal Society Open Science dx.doi.org/10.1098/rsos.140444

Carter, AJ; Marshall, HH; Heinsohn, R & Cowlishaw, G. 2014. Personality predicts the propensity for social learning in a wild primate. PeerJ, 2, e283.

Carter, AJ; Feeney, WE; Marshall, HH; Cowlishaw, G & Heinsohn, R. 2013. Animal personality: what are behavioural ecologists measuring? Biological Reviews, 88, 465-475.

Carter, AJ; Marshall, H; Heinsohn, R & Cowlishaw, G. 2012. Evaluating animal personalities: do observer assessments and experimental tests measure the same thing? Behavioral Ecology and Sociobiology, 66, 153-160.

Sih, A., Bell, A. M., Johnson, J. C. & Ziemba, R. E. 2004. Behavioral syndromes: an integrative overview. The Quarterly Review of Biology,79, 241–277.

*4) How were dental patterns of tooth eruption and wear observed? Were the baboons trapped and sedated? A brief background section on the history, management and handling of the population would be useful. Were their diets supplemented by humans in the past? If so, how was it done?*

We have added a subsection to the Materials and Methods that details the history of the population (subsection “History of the study population”). As specified in the Ethics section, all our protocols are independently approved by the ZSL Ethics Committee, adhere to the Guidelines for the Use of Animals in Behavioural Research and Teaching (Animal Behaviour 2003, 65:249–255), and meet the legal requirements of Namibia where the work was carried out.

*5) There was some confusion among the reviewers about how the data on grooming and dominance interactions were collected. Certainly the authors took pains to collect proximity data in a systematic manner that allowed them to compare individuals. However, it is less clear that this was the case for grooming and dominance-related interaction. The authors state that "Interaction data were recorded ad libitum by all observers at the project (n=6-7 in total) between dawn and dusk across all individuals as they moved continuously through the troop". Given that troops could be dispersed over 1 square km, was what they did equivalent to all occurrences sampling? Could the entire troop be observed at once and all instances of grooming and agonism recorded? Do the authors mean that all 6 or 7 observers collected data at the same time, working different parts of the troop? If so, what steps were taken to make sure coverage was even and that individuals were not sampled more than once over a short interval of time? Was the amount of time each individual was observed recorded and used to calculate rates? Was any other sort of correction made for variation in the observability of individuals? If not, this could have seriously biased the social networks they derived. Given that the ranges of grooming interactions and dominance interactions per initiator were so large (1-111 and 1-116, respectively), we suggest that it may be better to cite the means and ranges of total grooming and dominance interactions per subject rather than initiations.*

That the interaction data we collected was representative of actual rates of interaction was a concern of ours when designing our data collection and we apologise for the confusion introduced by the brevity of this section. We have now elaborated to clarify (Materials and Methods, subsection “Interaction networks”, first paragraph). No, unfortunately it is impossible to perform all-occurrences sampling of interactions, nor is it possible to record how much time observers spend in proximity to particular baboons to estimate sampling rates. However, as we explain in the amended manuscript, we believe that our sampling method was not biased because all observers were required to move throughout the troops and to monitor all group members throughout the day as part of their other data collection duties, and they thus regularly sampled the entire troop. Like the referees, we were also surprised by the range in the observations of interactions; however, on inspection of these, they made biological sense and we were satisfied they were representative: frequent groomers were females in consortship with males (and often the same males over months); infrequent groomers were juvenile males; frequent antagonists were males in consortship with females; infrequent antagonists were low ranking individuals. The patterns in the ad lib interactions data are also verified by the behavioural states observed during the focal individual proximity scans, i.e., some individuals were frequent groomers, while others were never observed grooming (across 94 individuals, the percentage of observations taken during grooming ranged from 0 to 35%, with 14 individuals never recorded grooming); moreover, females in consortship groomed more than all other individuals (Welch two-sample t-test: t = -3.13, df = 42.05, p = 0.003) while juvenile males groomed less than others (Welch two-sample t-test: t = 6.35, df = 91.97, p < 0.001). Finally, we now also specify how we sampled to avoid pseudoreplication (in the second paragraph of the aforementioned subsection).

*6) Please clarify how dominance networks are different from dominance ranks for readers that are less familiar with social network analysis. Please also discuss your use of undirected and directed networks. While OADA can be used with both directed and undirected networks, the directionality of the grooming and dominance networks (and lack of directionality of the proximity networks) needs to be discussed more fully. We suggest the authors explain (a) how directionality affects their interpretations of their results and (b) if undirected networks were attempted for grooming and aggression to see whether pairs that interact aggressively/ submissive frequently also appeared to have monitored each other frequently enough to pass on social information.*

We have elaborated on the difference between dominance networks and ranks (subsection “Interaction networks”, second paragraph) and directed and undirected networks (fourth paragraph). We were interested by the idea that we use undirected versions of the directed networks, and have now included these new analyses. Overall, they do not change the essence of our main findings, but we did find that (1) the undirected nearest neighbour network was better than the directed network, and (2) there was no difference between the directed and undirected versions of the interaction networks. As such, we discuss how directionality of the network affects information transmission (in the fourth paragraph of the aforementioned paragraph and in the first paragraph of the Discussion).

*7) Six or seven observers were used. Were interobserver reliability tests done? If so, how and what was the criterion for passing?*

Again, we apologise for the ambiguity introduced by our brevity. For the collection of the interaction data, there was a total of 7 observers, with 6 remaining onsite for the whole season, but only 1-4 attended each troop on any day. We have added the following text to clarify (subsection “Interaction networks”, first paragraph): “On any given day, 1-4 observers were present with each troop, from a total pool of 7 observers for the field season.” We have also added subheadings for both proximity and interaction data collection to further emphasise that different methods were used in each case, i.e., while the interaction data were collected by all observers, only one observer (MTT) collected most (99.7%) of the proximity data (the other 0.3% were performed by AJC during training). While we perform concurrent observations for interobserver reliability tests for data collected using focal observations, we do not perform such tests for the ad libitum data. While grooming is never ambiguous, some dominance interactions can be more subtle and difficult to identify correctly (such as during recruitment for coalition formation where it may not be clear which individual is being threatened by whom). New observers to the site thus receive intensive training at the beginning of the field season, and are tested by experienced observers until they consistently identify dominance interactions correctly. This usually takes two weeks. Observers are advised to collect dominance data only when they are sure of the interaction. We have added text to this effect to the manuscript (subsection “Interaction networks”, first paragraph).

*8) Figure 1, please provide median and range for number of scans per individual.*

We have now provided this information in the third paragraph of the subsection “Interaction networks”.

*9) The authors state that they hid the food when out of sight from the troops, but stood 15 m away from it with a video camera as the monkeys approached. What steps were taken to insure that the presence of observers at this point did not provide cues to the monkeys that a food patch was nearby? Cues of this nature could potentially pose a serious flaw in the design of the experiment, especially given that baboons catch on quickly to such things, and the experiments were replicated 25 times for each troop.*

This was a concern of ours, not least because we do not want the baboons to associate observers with providing food. We have amended the text for clarity and added the following to qualify our methodology (Materials and methods, subsection “Information diffusion experiments:, first paragraph): “To avoid the baboons observing the patch being created, the observer either (i) quickly scattered the kernels as she was walking or (ii) pretended to get something from her field backpack while scattering the kernels behind her bag. In all cases, the baboons did not see the kernels being placed. Furthermore, because the observer was present in the troops for many hours preceding and following these trials, the baboons did not associate the camera nor the waiting behaviour of the observer with the presence of the patches.”

*10) More than half the time the patch was found by the leading edge of the troop. Please discuss how this may have influenced the outcome of the study and the results indicating the importance of position in the proximity network. Are the most central animals disproportionately found in the leading edge? Would it be possible to control for the part of the group that happened to encounter the patch first-or run separate analysis for each condition? A brief description of the attributes of animals in different positions of an advancing group might help interpret the findings. How many times was the patch not found by the baboons?*

The reviewers’ queries above can be synthesised as the following two questions: (1) is the transmission of information affected by who discovers the patch and (2) is patch discovery dependent on individuals’ phenotypes? This is our current research focus! That is, we are now investigating whether particular (social and behavioural) phenotypes are more or less likely to generate information and, if so, whether individuals can pay attention to such information generators strategically to maximise social information acquisition. Whilst we have included information about the numbers of individuals who found a patch to demonstrate the diversity of patch discoverers (Materials and methods, subsection “Information diffusion experiments”, first paragraph), we are aware that this is not information about their phenotypes. Because this is a research topic for which we have been collecting data we do have some preliminary analyses to answer the reviewers’ queries. These suggest that older individuals are more likely to be on the periphery of the troop and higher ranking individuals tend to be at the front, though this latter relationship is not statistically significant. Perhaps surprisingly, the proportion of time individuals are found on the periphery or front of the troop does not predict whether or not they will find a patch, though again we stress that these analyses are preliminary. We have not yet analysed whether network and spatial positions are correlated. However, we would prefer not to present these results in the current manuscript. Our reasons for this are two-fold: (1) the analyses answer a different research question tangential to the main research questions, and (2) we are currently collecting a more robust dataset (pooled over 3 years) to answer this question and do not wish to present these limited analyses here.

*Did the 50 experiments include only those trials in which the patch was found?*

We now describe the numbers of failed trials in the text, together with a brief explanation of why the baboons failed to detect the patches in those cases (Materials and methods subsection “Information diffusion experiments”, first paragraph).

*11) The determination of social acquisition of information about the patch was via gazing of an individual toward another individual that was in a patch. Given that baboons constantly monitor each other's positions with gaze, were frequencies of gazing corrected for baseline rates of gazing at other individuals not in the patch? Was there a difference, e.g., in the duration of gazes, that would indicate the likelihood that information was actually acquired and that the gaze was not a routine monitoring of position only?*

This is a good point. We did discuss this possibility inasmuch as we were concerned whether we could be certain that an individual had acquired social information about the location of the patch. However, the routine monitoring of conspecific position tends to occur through rapid glances, rather than the prolonged gazes that we recorded. In addition, the vast majority of individuals approached the patch and stopped foraging to watch individuals in the patch (as is evident in the videos provided), which further indicates more than simple routine monitoring.

*12) Results subsection “Identifying phenotypic constraints on social information use”, last paragraph and elsewhere. What does 'average (median)' mean? Average (meaning mean) and median are the same?*

“Average” indicates the central tendency of data (e.g. mean, median, mode) and is not synonymous with mean, which is why we specified which average we used in all cases.

*13) In the Discussion, the authors should explicitly consider the implications of using only small and ephemeral food patches. The brief nature of their food patch experiments means that animals that acquired information about the location of the food patch (and perhaps even applied it, though unsuccessfully) did not have the opportunity to apply and exploit such information in the future. From an evolutionary perspective, however, social information can spread about more permanent food patches as well as ephemeral food patches, and there may be future opportunities for animals to return to the location of more permanent food patches and they may be able to apply the information acquired (about this new food patch) as well as successfully exploit it. This may even be a reason for low-ranking individuals to enter a depleted food patch (especially if visibility of patch details is somewhat limited without entering it) – to determine for themselves whether the food patch may be a permanent patch that they can return to at a later date.*

Again, the reviewers have highlighted a new research question raised by the findings of this study and for which we have now collected data to test: Does ephemeral information transmit through different social networks to those that transmit longer-lasting information? We did originally discuss this in our earlier versions of the manuscript, but removed the paragraph because of the length of the manuscript. We have now added this back to our Discussion on the encouragement of the reviewers’ interest in the idea.